# Fe/Mg/Fe Multilayer Composite Sheet Fabricated by Roll Cladding

**DOI:** 10.3390/ma15144732

**Published:** 2022-07-06

**Authors:** Daxin Ren, Yanhua Ma, Rencheng Zheng

**Affiliations:** 1School of Automotive Engineering, Dalian University of Technology, Dalian 116024, China; rendx@dlut.edu.cn; 2Tianjin Key Laboratory of Equipment Design and Manufacturing Technology, Tianjin 300072, China; rencheng.zheng@tju.edu.cn; 3School of Microelectronics, Dalian University of Technology, Dalian 116024, China; 4School of Mechanical Engineering, Tianjin University, Tianjin 300072, China

**Keywords:** cladding, magnesium, steel, microstructure, composites, hardness

## Abstract

A new multilayer composite sheet consisting of Fe/Mg/Fe was fabricated from galvanized steels and Mg alloy sheets via roll cladding. The clad steel improved the Mg surface hardness from HV 65 to HV 132. Bonding occurred as the reduction ratios increased up to over 10%. Investigation of the microstructure of the Mg/steel interface revealed a 5 μm- to 10 μm-thick transition layer between Mg and each steel sheet, consisting of Zn and an intermetallic compound (0.97Mg–0.03Zn). Zinc coating from the galvanized steel sheet improved the metallurgical bonding between Mg and Fe by forming new intermetallic phases.

## 1. Introduction

Lightweight materials have increasingly gained attention due to their desired properties, such as high strength and low density. Magnesium alloys are widely used in the automotive industry. However, magnesium alloys have some properties, such as poor corrosion resistance, low wear resistance, and high chemical reactivity, limiting their use in the industry [1,2]. Conversely, materials with such properties do not satisfy the requirements for advanced applications. Numerous methods have been employed to enhance alloy properties. Plasma electrolytic oxidation is an attractive surface process for magnesium alloys, and the surface of a magnesium alloy is converted into a hard ceramic coating [3,4]. Coatings based on chitosan and bioactive glass particles fabricated by electrophoretic deposition pretreated magnesium alloys have promising potential to suppress the substrate corrosion and additionally incorporate bioactivity [5,6]. Spray coatings are used to improve the corrosion resistance of magnesium alloys [7,8] However, such surface-treatment methods are performed after casting or deformation, requiring additional manufacturing processes [9]. Thus, alternative cost-effective approaches must be developed.

Roll-clad composites exhibit better physical and chemical properties than other traditional materials [10,11]. Roll-clad composites are produced by rolling Mg alloy sheets without additional surface-treatment processes. Steels and Al alloys are significantly used to produce roll-clad composite structures. Mg/Al and Mg/Al/stainless steel composite sheets have been successfully fabricated using the roll-cladding process [12,13,14,15], but only a few studies have focused on fabricating Fe/Mg/Fe multilayer composites. Fe/Mg/Fe multilayer composites can potentially increase surface hardness, resistance corrosion, and magnetic property, useful in engineering applications. However, Mg alloys and steels are difficult to roll or weld because of their different large properties [16]. The boiling temperature of Mg alloys is considerably lower than the melting point of steel [17]. In addition, Mg hardly reacts with Fe, and the intersolubility between the two materials is negligible at ambient pressure [18]. These differences severely limit the metallurgical bonding between steel and Mg alloys.

Galvanized steel (GS) offers higher wear and corrosion resistance than most Mg alloys. In the present study, an improved technique for fabricating Fe-clad Mg alloy sheets was proposed. A three-ply composite sheet composed of GS/Mg/GS was produced from GS and Mg alloy (AZ31) sheets through roll cladding. The interfacial microstructures and bonding mechanism of the composite were then investigated.

## 2. Materials and Methods

A commercial AZ31 Mg alloy (Mg–2.93Al–0.67Zn–0.43Mn–0.08Si) and galvanized steel (Fe–0.04C–0.2Mn–0.06Al–0.02Cu) were used in the experiments. The AZ31 Mg alloy sheet was rolled to 1.7 mm before roll cladding. 0.1 mm thick steel sheets with different thick Zn coating were used, and the thickness was 2 μm and 20 μm, respectively. Steel sheets were cut into 300 × 100 mm pieces; this size was the same as the Mg alloy sheet. Before roll cladding, the Mg alloy and galvanized steel sheets were stacked in a Fe–Mg–Fe sequence and heated up to 400 °C for 10 min. Figure 1 shows the schematic diagram of the roll-cladding process. Two roll mill was used in the experiment. The rolling velocity and diameter of rolls are 250 mm and 42 r/min. Different nominal deformation was set to investigate the bonding properties. After the rolling process, the roll-clad composites were cooled in air.

The morphology across the clad interface of fabricated Fe-clad Mg composite was characterized using a scanning electronic microscope (SEM) equipped (Carl Zeiss SUPRA55, Oberkochen, Germany). Elemental mapping was performed using an electron probe microanalyzer (JEOL-8530F Plus, Japan). The phase was detected by X-ray diffraction (Empyrean, Netherlands), and the microhardness of the sheet was tested at a load of 100 g for 15 s.

## 3. Results and Discussion

Four reduction ratios (5%, 10%, 15%, and 20%) were tested in the roll-clad process, and the result is shown in Table 1. When the reduction ratio is 5%, the Mg–Fe interface cracks after the roll-clad process. Strong bonding occurs as the reduction ratios increase up to over 10%. The high chemical reactivity of Mg alloy always results in oxide film on the surface, and the film is a barrier for dissimilar sheets in the roll bonding process. Zn and Mg can intermetallic react at relatively low temperatures, so the bonding threshold of the reduction ratio is relatively low compared to other roll-clad composites.

Wear resistance is not always related directly to hardness because of many wear mechanisms; however, hardness remains a helpful indicator. The microhardness of the Mg sheet and the composite sheet surface was measured after roll cladding. The microhardness of the AZ31 substrate is approximately HV 65, whereas that for the composite sheet surface is relatively higher (HV 132 to HV 138). A substantial increase in microhardness enhances wear resistance. Moreover, GS is a commonly used corrosion-resistant material that can increase corrosion resistance. The magnetic property of the composite sheet can also facilitate easy transport and assembly. The mechanical properties of the composite sheet contribute to many potential engineering applications.

After the rolling process, the produced multilayer sheets with different thicknesses of Zn coating showed no visible difference, as shown in the cross-section image of the composite sheet in Figure 2. As the hardness of Fe is higher than that of Mg, deformation mainly occurred in the softer AZ31 sheets. As a result, the thickness of the steel sheet nearly remained at 0.1 mm during hot roll cladding. No obvious defects were observed along with the AZ31/steel clad interface.

Mg/Fe interfaces with varying thicknesses of Zn coating exhibited different microstructures, as shown in Figure 3a,b. Transition layers formed at the interface exhibited different microstructural characteristics than the AZ31 substrate and clad steel sheet. Interfaces with 2 μm-thick Zn coating showed a thin transition layer with a width of 5 μm to 10 μm, whereas the thickness of the transition layer with 20 μm-thick Zn coating was up to 33 μm. Moreover, the thicker Zn coating generated complex transition layers. Two layers with different microstructural characteristics are shown in Figure 3b, in which layer 2 (L2) exhibited etch-pits, and layer 1 (L1) exhibited lamellar structures.

EDX was used to analyze the composition of different regions across the transition layers in Figure 3. As shown in Table 2, Zn contents in both transition layers were higher than in the AZ31 Mg alloy, which indicated that Zn from the GS can penetrate the AZ31 substrate during the rolling process. However, the maximum Zn content in the transition layer was only 4.2% when the 2 μm-thick Zn coating was used. In this case, the Zn coating is almost wholly diffused into the AZ31 substrate, and the amount is not sufficient to form large Zn-rich structures. The complex transition layer is shown in Figure 3b has a composition gradient from the AZ31 side to the steel side. The average contents of Mg and Zn are 58% and 35% in layer 1 and changed to 34% and 63% in layer 2, respectively. Based on the Mg–Zn phase diagram and the microstructure observation, layer 1 can be inferred to have eutectic structures of α-Mg and MgZn. The Zn/Mg (at.%) ratio was approximately 2:1 in layer 2. Thus, layer 2 could be mainly composed of MgZn_2_ combined with other intermetallic compounds (IMCs). The above finding is very similar to an Mg/Al welded joint with a Zn interlayer [19,20]. As reported in the literature, Mg–Zn intermetallic compounds, such as MgZn and MgZn_2_, are characterized by high brittleness, which reduces the strength of the welded joint [21,22]. When the 20 μm-thick Zn coating was used, the excessive Zn resulted in a thick transition layer composed of Mg–Zn intermetallic compounds, which could severely affect the mechanical properties. Hence, the 2 μm-thick Zn coating was the most suitable for Fe/Mg/Fe multilayer sheet roll cladding based on the microstructures.

To elucidate the elemental diffusion and formation mechanism of the microstructure in the roll cladding process using the 2 μm-thick Zn coating, elemental mapping of the interface was performed to determine the chemical composition (Figure 4). The elemental distribution showed composition gradients of Mg, Fe, and Zn at the interface. Results indicated that elemental diffusion occurred in each layer. Zn was concentrated at the interface, and the width was nearly the same as the width of the transition layer in Figure 3a. The Mg and Fe contents gradually decreased in the transition layer compared with those in the AZ31 substrate and steel sheet. Elemental analysis showed that Zn is vital in bonding steel and AZ31. Furthermore, the intersolubility between Mg and steel was negligible. Thus, the transition layer was formed by either an Mg–Zn reaction or a Fe–Zn reaction. The detailed mechanism is discussed in the subsequent paragraphs.

The roll-clad composite sheet was peeled to investigate the bonding characteristics. The peeled-off specimen from the AZ31/steel bonding interface because of the weak multilayer composite sheet. Figure 5a,b shows the peel-fracture surfaces of the steel side and AZ31 side, respectively. Both fracture surfaces exhibited large cleavage planes, which suggests the occurrence of a typical brittle fracture during the peel test.

The XRD patterns of the fracture surfaces are shown in Figure 5c,d. Fe was present in the fracture surface of the steel side. However, no Fe–Zn compounds were detected. Meanwhile, Zn was detected on both sides, which indicated that the Zn coating did not completely melt or react with Mg due to the high cooling speed. However, Mg was not detected on the fracture surface of the AZ31 side, and only the 0.97Mg–0.03Zn phase was observed. The appearance of the new phase indicated that during the roll cladding process, the Zn coating diffused into the AZ31 and reacted with Mg under the high heat and pressure conditions. The solid solubility limit of Zn in Mg is approximately 2% and implies that the 0.97Mg–0.03Zn phases can precipitate from the supersaturated solid during the cooling process. Intermetallic compounds are generally characterized by brittleness or low ductility. The observed composite sheet fractures from the 0.97Mg–0.03Zn phase can be attributed to the abovementioned characteristics. Similar results were reported in the deformation of Mg alloys, particularly in welding Mg alloy to steel via solid-phase friction-stir welding [23,24,25]. The XRD results showed that the Zn coating from GS improved the metallurgical bonding of AZ31 and steel. The two materials were bonded by the transition layer composed of Zn and the 0.97Mg–0.03Zn intermetallic compound.

## 4. Conclusions

GSs were roll-clad on the AZ31 Mg alloy sheet surface, and a new multilayer composite sheet composed of Fe/Mg/Fe was successfully produced. Bonding occurred as the reduction ratios increased up to over 10%. Results revealed that clad Fe significantly improved the surface hardness of the Mg alloy sheet from HV 65 to HV 132. A transition layer that consists of Zn and a 0.97Mg–0.03Zn IMC was formed at the steel/AZ31 interface. This transition layer bonds the different materials. Furthermore, the Zn coating from GS improved the metallurgical bond between Mg and Fe.

## Figures and Tables

**Figure 1 materials-15-04732-f001:**
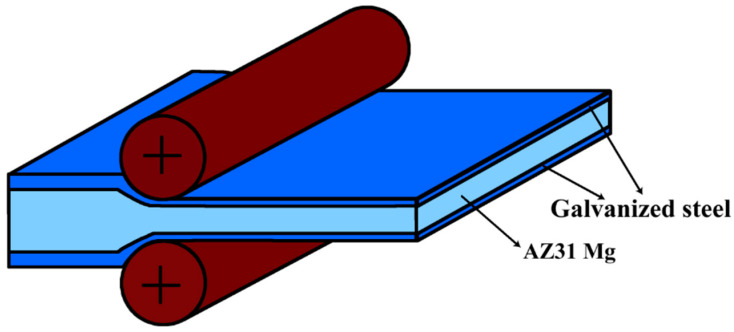
Schematic of the roll cladding of a multilayer composite sheet.

**Figure 2 materials-15-04732-f002:**
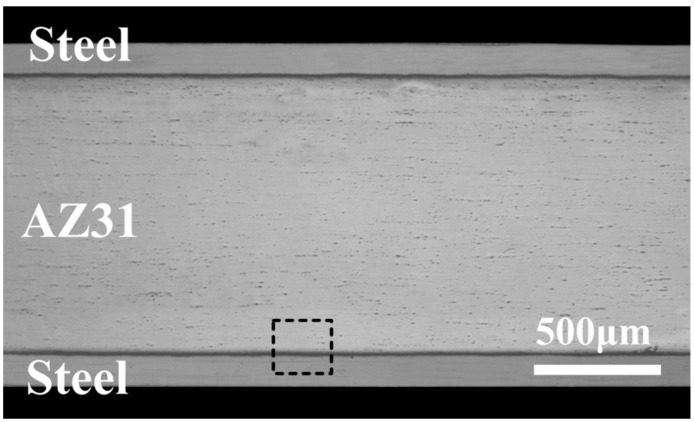
Cross-section of the multilayer sheet.

**Figure 3 materials-15-04732-f003:**
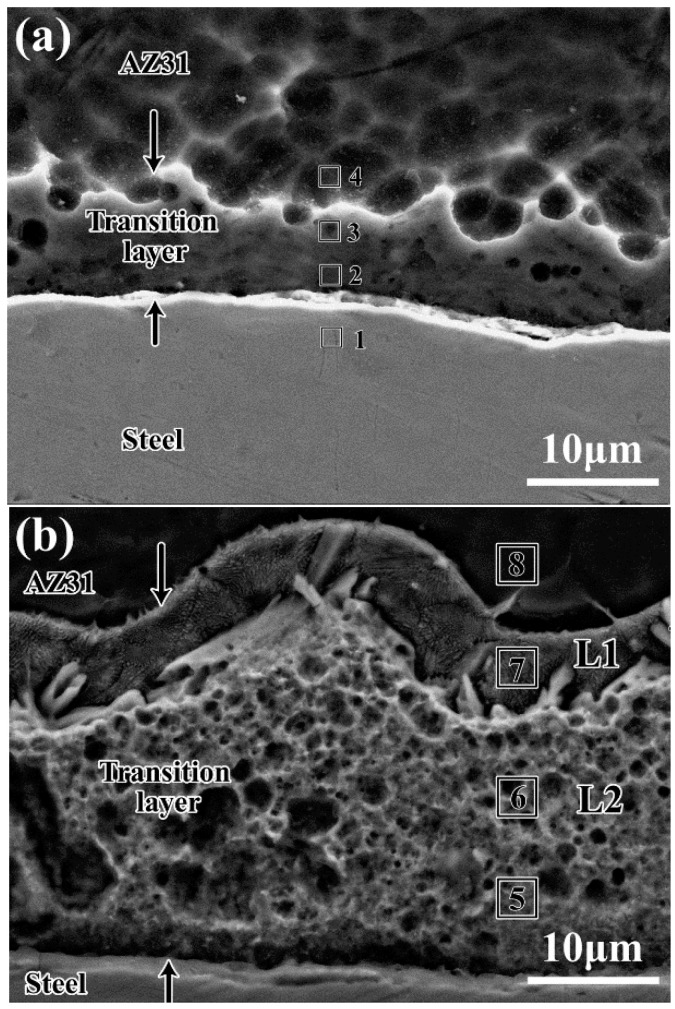
Microstructures of the bonding interfaces depicting the (**a**) 2 μm-thick zinc coating, and (**b**) 20 μm-thick zinc coating (Numbers 1–8 are the test positions of Table 2).

**Figure 4 materials-15-04732-f004:**
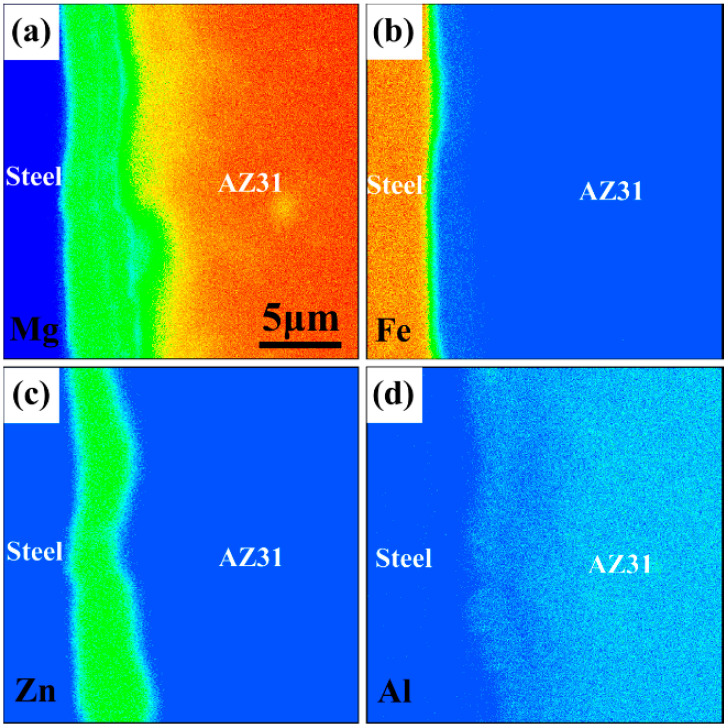
Map analysis of the interface: (**a**) Mg, (**b**) Fe, (**c**) Zn, and (**d**) Al.

**Figure 5 materials-15-04732-f005:**
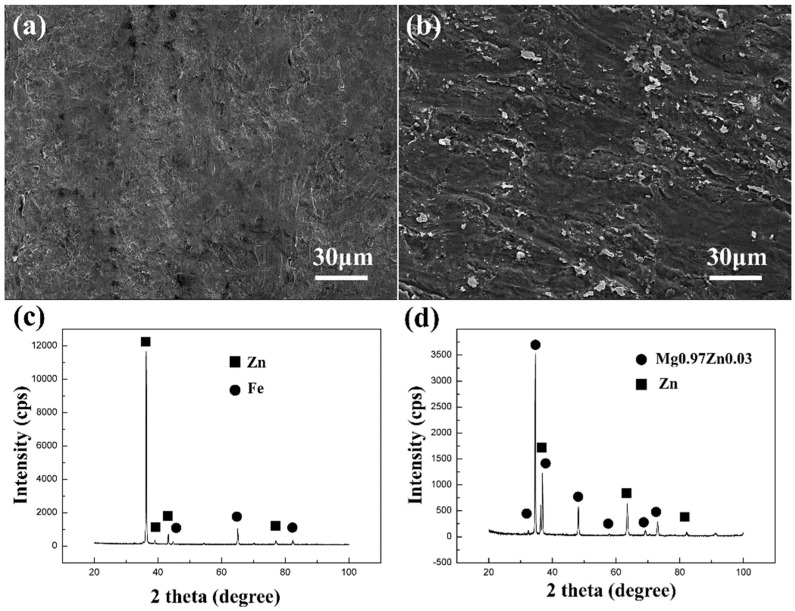
(**a**,**b**) Fracture surfaces of the steel side and of the AZ31 side. (**c**,**d**) XRD spectra of the steel side and of the AZ31 side (Zn: ICCD card M010870713; Fe: ICCD card N010851410; Mg0.97Zn0.03: ICCD card N030654596;).

**Table 1 materials-15-04732-t001:** Roll-clad results of Fe/Mg/Fe composite.

Rolling Draft	Fe/Mg/Fe Composite
5%	Not bonded
10%	bonded
15%	bonded
20%	bonded

**Table 2 materials-15-04732-t002:** Element composition of different regions in Figure 3, in at pct.

Elements	R-1	R-2	R-3	R-4	R-5	R-6	R-7	R-8
Mg	0	92.6	93.7	96.1	32.6	35.4	58.6	95.2
Al	0.4	2.7	1.9	2.7	2.9	2.1	5.8	2.6
Zn	0.2	4.4	4.2	1.2	64.1	62.1	35.4	2.2
Fe	99.3	0.3	0.2	0	0.4	0.4	0.2	0

## Data Availability

Data sharing does not apply to this article as no new data were created or analyzed in this study.

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
