# Peer review of "Fe/Mg/Fe Multilayer Composite Sheet Fabricated by Roll Cladding"

_materials, 2022, doi:10.3390/ma15144732_

Round 1

Reviewer 1 Report

please look on this issue https://aip.scitation.org/doi/abs/10.1063/1.5112581

Reviewer 2 Report

Regarding that the materials-1810125 has been resubmitted as "a communication paper", it is suitable for publication in its present form.

Reviewer 3 Report

I think the article is okay for publication 

This manuscript is a resubmission of an earlier submission. The following is a list of the peer review reports and author responses from that submission.

Round 1

Reviewer 1 Report

line 54 - mm instead mm2

please show thickness distribution between steel and Mg-alloy depending on rolling reduction: initial - 0.15 mm, after rolling "nearly remained constant at 0.1 mm during hot roll cladding". It means 33% of reduction?

what about mechanical properties of the composite?

Author Response

(1) line 54 - mm instead mm2

This mistake has been corrected.

(2) please show thickness distribution between steel and Mg-alloy depending on rolling reduction: initial - 0.15 mm, after rolling "nearly remained constant at 0.1 mm during hot roll cladding". It means 33% of reduction?

The initial thickness was 0.1 mm. The thickness in the manuscript was not accurate and has been revised.

(3) what about mechanical properties of the composite?

In this manuscript, we mainly put forward a new idea. The main purpose of this process was to improve the surface hardness of magnesium alloy. The mechanical properties, such as tensile strength and bending strength, depending on the subsequent heat treatment process and the properties of the steel sheet. Theoretically, galvanized steel sheets, including many kinds of high-strength steel and ultra-high-strength steel, can be used. In this paper, we used low-carbon steel to realize the proposed idea. Other types of steel sheets and mechanical properties will be researched in the future.

Reviewer 2 Report

The present work evaluates a multilayer composite sheet consisting of Fe/Mg/Fe sheets via roll cladding. While the paper reports some experimental findings, it lacks novelty in data analysis and discussion. Moreover, the present data are too low to be published in a high-quality journal. It is more suitable for a conference paper rather than a scientific journal. A more comprehensive evaluation is required for preparing a scientific journal. Other issues are:

1.      The authors can add some quantitative results in the abstract.

2.      How do the authors characterize the bond strength? Did the authors perform any peeling test to show the bond strength? The peeling force versus peeling distance can be plotted.

3.      The authors should distinguish the bonding and un-bonding region in the peeled-off surfaces. 

4.      In the conclusion and abstract, the authors mentioned the wear, corrosion, and magnetic properties, but they have not evaluated these properties.

Author Response

  1. The authors can add some quantitative results in the abstract.
  • Relevant contents have been supplemented in the abstract.
  1. How do the authors characterize the bond strength? Did the authors perform any peeling test to show the bond strength? The peeling force versus peeling distance can be plotted.
  • Unfortunately, we have not been able to test the peel strength. The sheets were bonded by Mg-Zn IMCs at the interface, but the IMC are brittle and the peel fracture was easy to occur. Besides, Magnesium alloys have poor plasticity at room temperature and are difficult to bend. So, we could not find a good way to quantitatively test the peel strength.
  1. The authors should distinguish the bonding and un-bonding region in the peeled-off surfaces. 
  • Bonding occurred as the reduction ratios increase up to over 10%, and in this condition, all the surfaces were bonded. The region shown in Figure 5 is all bonding regions.
  1. In the conclusion and abstract, the authors mentioned the wear, corrosion, and magnetic properties, but they have not evaluated these properties.
  • In this manuscript, we mainly put forward a new idea. The improved corrosion and wear resistance are potential advantages due to many previous studies in the field of materials. These properties depend on the properties of the steel sheet and the subsequent heat-treatment process. We will conduct systematic experiments in the future.

Author Response

1: line 23 24: what does poor corrosion means? Please complete the sentence.

  • The sentence has been revised.

2: in the key words, all first letters should be either capital or not.

  • The mistakes have been corrected in the manuscript.

3: the introduction is brief yet sufficient. The only thing that I would be interested to know is why GS-Mg-GS scheme has been selected and not any other possibility like Mg/Fe/Mg etc.

  • Mg/Fe/Mg structure mentioned by the reviewer is also very meaningful, and I think it can increase the strength of the Mg sheet. Magnesium alloy sheet surface has a relatively low hardness and corrosion resistance compared to steel, and this greatly limits the application of magnesium alloys. So, We put forward a new structure of GS/Mg/GS to improve these properties in this paper.

4: I suggest enhancing the introduction of the reasons behind the selection of GS. What other materials can be used in regard to this type of applications?

It is difficult for magnesium and steel to metallurgical react, so a third element was needed to realize the metallurgical bonding of the two sheets. Zn can be well metallurgical react with both magnesium and steel. In the previous work, we tried to put zinc foil between the magnesium alloy sheet and steel sheet, but the experimental results were not satisfactory. And finally, Finally, we chose galvanized steel, which could make the effect of composite plate manufacturing better.

5: Line 61: SEM can tell you the morphology and not the microstructure.

  • The mistakes have been corrected in the manuscript.

6: Please add the names of the equipment and the conditions for the measurements in the characterization details of the experimental parts… line 61 to 64.

  • The relevant information has been added to the manuscript.

7: the results and discussion heading means that the results should be elaborated and a discussion must be provided for the achieved results. It seems that the authors have written the article in rusha and thus they have not elaborated the reasons behind these results and as previously said the characterization conditions. It is highly recommended for the acceptance of the article that the authors discuss results.

  • In this manuscript, we mainly put forward a new idea to produce GS/Mg/GS composite sheet. This paper reports some experimental findings and can be regarded as rapid communication. So, it lacks comprehensive analysis and discussion.

8: in the keywords and in the results, the authors have emphasized on the wear resistance. this is an empirical judgement based on the hardness results. It is not a thing to be assumed so, please provide adhesion and wear testing results. Hardness can enhance due to increase içnt he stresses at the interface.

  • The improved corrosion and wear resistance are potential advantages due to many previous studies in the field of materials. These properties depend on the properties of the steel sheet and the subsequent heat-treatment process. We will conduct systematic experiments in the future. Relevant improper statements have been modified in the manuscript.

9: what are the advantages and disadvantages of the porosity observed in the figure 3b. how it can be avoided. Please discuss the results for better understanding and develop a comparison with the literature. This is recommended for all the results.

  • Porosity couldn’t be seen after the specimens were polished. The porosity seen in Fig. 3b was caused by etching

10: In the EDX measurements, did the quatification of O and C were taken? If not, please explain why. I think it is important to be added. But keep in mind that the C and O measurements in EDX are not accurate. It is recommended that the authors use WDS for more reliable quantification.

  • I agree with the reviewer. Zn at the interface was easy to oxidize after peeling, so the O element was not measured in the EDX.

11: Figure 5 c and d are not readable.

  • The picture has been re-edited.

12: please add the ICCD card numbers for XRD results

  • ICCD card numbers have been added

13: for suture works, heat treatment studies of these developed layers can be a good addition to the literature.

  • I agree with the reviewer. Heat treatment can promote the reaction and diffusion of the elements at the interface, which is helpful to improve the strength of the interface. Systematic and comprehensive experiments will be done in the future.

14: please add author contributions

  • The author's contributions have been added to the manuscript.

15: References are missing dois

  • When writing the paper, we referred to some published papers in Materials, and the references in the papers did not have DOI. And the references did not require DOI when we submitted the paper before.

Reviewer 4 Report

The paper treats an up-to-date problem the effect of rolling parameters (mainly deformation) on the quality of the joining area in the steel/AZ31/seel multi-layered materials. In the paper I found some inaccuracies that should be explained and corrected:

1. Each one (two) of the quoted references should be discussed individually and demonstrate their significance to the work. It is not necessary used four or even seven references in one bracket: [3-9], [12-15], [16-18].

2. Line 35: “…but only a few studies have focused on fabricating Fe/Mg/Fe multilayer composites.” The references used are sufficient for the paper's issues clarification. However, Authors should add some literature at the end of the sentence.

3. Line 51: “The AZ31 Mg alloy sheet was rolled to 1.7 mm before roll cladding.” What was an initial thickness of the sheet?

4. There are no rolling parameters: type of rolling mill, rolling velocity, diameter of rolls

5. Table 1: is Roll cladding condition; better is deformation or rolling draft

6. What was a final thickness of the particular layers for different deformation?

7. Fig. 3. What was an effect of the deformation on the thickness of the transition layer?

8. Figs. 3-5. For which variant of the results are shown?

9. Line 149, is ZA31, should be AZ31

10. Conclusions are not proper. The sentence: “Results revealed that clad Fe significantly improved the surface properties of Mg alloys, including wear resistance, corrosion resistance, and magnetic property.” There are no results of the tests.

Author Response

  • Each one (two) of the quoted references should be discussed individually and demonstrate their significance to the work. It is not necessary used four or even seven references in one bracket: [3-9], [12-15], [16-18].

Different technologies have been discussed individually

  • Line 35: “…but only a few studies have focused on fabricating Fe/Mg/Fe multilayer composites.” The references used are sufficient for the paper's issues clarification. However, Authors should add some literature at the end of the sentence.

Literature has been added to state the issues.

  • Line 51: “The AZ31 Mg alloy sheet was rolled to 1.7 mm before roll cladding.” What was an initial thickness of the sheet?

The initial thickness of the AZ31 sheet was 2.0mm.

  • There are no rolling parameters: type of rolling mill, rolling velocity, diameter of rolls

Two roll mill was used in the experiment. The rolling velocity and diameter of rolls are 250mm and 42 r/min. This information has been supplemented in the manuscript.

  • Table 1: is Roll cladding condition; better is deformation or rolling draft.

The mistakes have been corrected in the manuscript.

  • What was a final thickness of the particular layers for different deformation?

The final thickness was 1.76 mm, 1.67 mm, 1.57 mm, and 1.48 mm, respectively.

  • 3. What was an effect of the deformation on the thickness of the transition layer? Thickness of transition layer

The thickness of the transition layer had no obvious change under the same temperature and rolling velocity. Rolling time was short for the reaction of Mg and Zn, so the transition layer could not thicken and deformation.

  • 3-5. For which variant of the results are shown?

The results of a 20% rolling draft were shown in these figures.

  • Line 149, is ZA31, should be AZ31

The mistakes have been corrected in the manuscript.

  • Conclusions are not proper. The sentence: “Results revealed that clad Fe significantly improved the surface properties of Mg alloys, including wear resistance, corrosion resistance, and magnetic property.” There are no results of the tests.

In this manuscript, we mainly put forward a new idea. The improved corrosion and wear resistance are potential advantages due to many previous studies in the field of materials. These properties depend on the properties of the steel sheet and the subsequent heat-treatment process. We will conduct systematic experiments in the future. We will conduct systematic experiments in the future. Relevant improper statements have been modified in the manuscript.

Round 2

Reviewer 2 Report

The authors have responded to my additional issues in my review report. However, the main drawback of the manuscripts has not been addressed yet. I think the results are too low, and the paper is too short to be published as a regular paper. The article presents just four figures that explain the experimental findings. Therefore, I cannot accept it as a regular paper. It is more suitable to be published as a technical or short communication paper.

Reviewer 3 Report

I would say that the authors reply to the comment of adding discussion is not appropriate

They MUST discuss the results. The authors want to oublish this work as a research article and not as communication.

After these modifications, the work can be accepted 

Reviewer 4 Report

The paper is ready for publication.